# Structures of UBA6 explain its dual specificity for ubiquitin and FAT10

Ngoc Truongvan[1,2,5], Shurong Li[1,5], Mohit Misra[1,3,4], Monika Kuhn[1] & Hermann Schindelin [1]✉

The covalent modification of target proteins with ubiquitin or ubiquitin-like modifiers is initiated by E1 activating enzymes, which typically transfer a single modifier onto cognate conjugating enzymes. UBA6 is an unusual E1 since it activates two highly distinct modifiers, ubiquitin and FAT10. Here, we report crystal structures of UBA6 in complex with either ATP or FAT10. In the UBA6-FAT10 complex, the C-terminal domain of FAT10 binds to where ubiquitin resides in the UBA1-ubiquitin complex, however, a switch element ensures the alternate recruitment of either modifier. Simultaneously, the N-terminal domain of FAT10 interacts with the 3-helix bundle of UBA6. Site-directed mutagenesis identifies residues permitting the selective activation of either ubiquitin or FAT10. These results pave the way for studies investigating the activation of either modifier by UBA6 in physiological and pathophysiological settings.

The posttranslational modification of target proteins with ubiquitin alters their structure, function, and/or localization[1]. Ubiquitylation is carried out by a sequential enzymatic cascade of E1 activating enzymes, E2 conjugating enzymes and E3 ligating enzymes. In the presence of Mg-ATP, E1 catalyzes the acyl-adenylation of ubiquitin, initially forming a ubiquitin-AMP adduct. Subsequently, the E1 catalytic cysteine attacks the ubiquitin-AMP intermediate to form a thioester bond between the cysteine and the C-terminal glycine of ubiquitin, resulting in the formation of a covalent E1-ubiquitin adduct. Ubiquitin is then transferred to the active site cysteine of the E2 in a trans-thioesterification reaction. Finally, E3 enzymes catalyze the ligation of ubiquitin to their target substrates[2,3].

For many years UBA1 was thought to be the only E1 that activates ubiquitin, until in 2007 a second ubiquitin activating enzyme was discovered. This protein, referred to as UBA6, is present only in vertebrates and sea urchins and shares 40% sequence identity with UBA1 (Supplementary Fig. 1a). Furthermore, UBA6 is an unusual E1 enzyme as it activates both ubiquitin and the ubiquitin-like protein (Ubl) FAT10[4–6]. Different lines of evidence suggest that UBA6 does not simply represent a backup system for UBA1: (1) So far UBA6 and USE1 (also referred to as UBE2Z) are the only E1 and E2 which have been identified to be involved in FAT10ylation[5]. (2) UBA1 and UBA6 display distinct E2 selectivities, which partially depends on their C-terminal, E2-recruiting ubiquitin fold domains. These distinct E2 preferences permit the two E1 enzymes to direct ubiquitin to distinct subsets of E3 enzymes and consequently substrates. (3) UBA1 and UBA6 function in spatially distinct ways with ~99% of all ubiquitylation events being initiated by UBA1[7]. Although UBA6 is widely expressed in human tissues and cell lines, its expression is ten-fold lower compared to that of UBA1 in various cell lines[5], however, UBA6 expression can be upregulated, e.g. during dendritic cell maturation or hyperthermic stress[8,9].

Interestingly, UBA6 was found to activate ubiquitin and FAT10 both in vitro and in vivo[4–6]. FAT10 (also known as ubiquitin D), a member of the UBL family[10], is a two-domain protein with each domain folding into the β-grasp architecture also observed for ubiquitin[11]. FAT10 is expressed in mature dendritic cells and B cells, but it is also induced by the proinflammatory cytokines γ-interferon and tumor necrosis factor α in cells derived from various tissues. FAT10 targets

[1]Institute of Structural Biology, Rudolf Virchow Center for Integrative and Translational Bioimaging, University of Würzburg, Josef-Schneider-Straße 2, 97080 Würzburg, Germany. [2]Department of Molecular Oncology, Moffitt Cancer Center, Tampa, FL 33612, USA. [3]Institute of Biochemistry II, Goethe University Faculty of Medicine, Theodor-Stern-Kai 7, 60590 Frankfurt am Main, Germany. [4]Buchmann Institute for Molecular Life Sciences, Goethe University Frankfurt, Riedberg Campus, Max-von-Laue-Strasse 15, 60438 Frankfurt am Main, Germany. [5]These authors contributed equally: Ngoc Truongvan, Shurong Li. ✉e-mail: hermann.schindelin@virchow.uni-wuerzburg.de

proteins for rapid proteasomal degradation, with FAT10, in contrast to ubiquitin, being also degraded by the proteasome along with its substrate[11]. While ubiquitin can be efficiently transferred to many E2 enzymes by both UBA1 and UBA6, FAT10 exhibits a considerably higher selectivity since it is charged to USE1 only and USE1 solely accepts this modifier from UBA6[5]. Moreover, the higher selectivity of FAT10ylation towards USE1 is also reflected in the auto-FAT10ylation of USE1 in cis[12], which only takes place when the FAT10 C-terminal domain is present[11]. Parkin has been recently identified as the first E3 enzyme of FAT10[13].

Since ubiquitylation and FAT10ylation are involved in multiple cellular processes, it is not surprising that malfunctions in one or more components of this system lead to a variety of diseases[14]. UBA6-mediated proteasomal degradation was reported to be involved in brain-associated physiological and pathophysiological states in mice[15,16]. Interestingly, UBA6 was found to be overexpressed in human brains from patients with Alzheimer's disease[17]. The tumor suppressor protein p53 is a FAT10 substrate and a double-negative regulation of FAT10 and p53 was observed to be critical in the control of tumorigenesis[18], which is in line with the overexpression of FAT10 in many cancer cell types[18,19]. In hepatocellular carcinoma cells (HCCs), FAT10 is overexpressed and FAT10ylation facilitates degradation of the Wnt-induced secreted protein-1 (WISP1) by the proteasome. As WISP1 suppresses proliferation of hepatocellular carcinoma cells, its FAT10ylation-mediated degradation promotes tumor progression[20].

Due to the involvement of UBA6-mediated ubiquitylation and FAT10ylation in various cellular processes in both physiological and pathophysiological settings, it is imperative to study the underlying processes and decipher the molecular basis for the dual specificity of UBA6. Ultimately, targeting of UBA6 with specific inhibitors would not only permit the analysis of the consequence of blocking the entire FAT10ylation pathway and those ubiquitylation events which are catalyzed by UBA6, but to also develop alternative therapeutic approaches. Here, we report structures of UBA6, in complex with ATP and FAT10, and identify residues which selectively interfere with either the activation of ubiquitin or FAT10.

## Results

### Overall structure of UBA6

We initially determined the crystal structure of UBA6 in complex with ATP from an orthorhombic crystal form (space group $P2_122_1$) at a resolution of 3.3 Å (Table 1). Subsequently, crystals belonging to space group C2 became available, which diffracted anisotropically to 2.7/3.8 Å and, due to the higher resolution, will be described here. Like the $P2_122_1$ crystals, the C2 crystals contain two molecules in the asymmetric unit, which are well defined in the electron density maps

## Table 1 | Data collection statistics and refinement parameters

| Data collection | | | | |
|---|---|---|---|---|
| | UBA6-ATP | UBA6-ATP | Uba1-ATP | UBA6$_{chim}$-FAT10 |
| Space group | $P2_122_1$ | C2 | C222$_1$ | P1 |
| Cell dimensions | | | | |
| a, b, c (Å) | 104.8, 144,3, 206.7 | 123.9, 113.7, 183.5 | 107.72, 118.06, 196.65 | 93.5, 93.6, 109.5 |
| α, β, γ (°) | 90, 90, 90 | 90, 96.5, 90 | 90, 90, 90 | 70.5, 88.5, 74.6 |
| Resolution limits (Å, if anisotropic, best/worst high resolution) | 50.17–3.32 | 48.25–2.71/3.83 | 47.25–1.72/2.24 | 46.09-3.27/4.45 |
| $R_{merge}$[a] | 0.376 (4.031) | 0.294 (2.570) | 0.101 (0.964) | 0.202 (1.09) |
| $R_{pim}$[b] | 0.118 (1.360) | 0.123 (1.158) | 0.043 (0.414) | 0.120 (0.640) |
| CC$_{1/2}$ | 0.996 (0.408) | 0.9847 (0.4086) | 0.9988 (0.7039) | 0.988 (0.401) |
| $<I/\sigma I>$[c] | 5.2 (0.6) | 6.0 (0.9) | 10.6 (1.9) | 5.3 (1.3) |
| Overall spherical completeness (highest resolution shell) | 0.993 (0.947) | 0.725 / 0.218 | 0.693 (0.126) | 0.591 (0.094) |
| Overall elliptical completeness (highest resolution shell) | n.a. | 0.948 (0.774) | 0.944 (0.693) | 0.859 (0.534) |
| Redundancy | 10.8 (9.3) | 6.66 (5.56) | 6.65 (6.34) | 3.84 (3.90) |
| Wilson B-factor (Å$^2$) | 97.4 | 70.3 | 34.1 | 112.9 |
| Refinement | | | | |
| No. Reflections (work/test set) | | 48,998/2486 | 87,255/4617 | 30,793/875 |
| [d]$R_{work}$ (highest shell) | | 0.232 (0.260) | 0.166 (0.255) | 0.216 (0.304) |
| [e]$R_{free}$ (highest shell) | | 0.264 (0.266) | 0.207 (0.277) | 0.239 (0.358) |
| [f]Ramachandran statistics (%) | | 97.3/2.6/0.1 | 96.4/3.5/0.1 | 96.4/2.7/0.9 |
| Overall B-factor (Å$^2$) | | 75.8 | 44.2 | 111.3 |
| Data precision index (Å) | | 0.39 | 0.19 | 0.36 |
| RMS deviations in | | | | |
| Bond lengths (Å) | | 0.004 | 0.006 | 0.005 |
| Bond angles (°) | | 0.703 | 0.822 | 0.757 |
| Torsion angles (°) | | 16.76 | 15.09 | 17.32 |
| Planar groups (Å) | | 0.005 | 0.005 | 0.008 |

*n.a.* not applicable.

[a]$R_{sym} = \Sigma_{hkl}\Sigma_i|I_i - <I>|/ \Sigma_{hkl}\Sigma_i I_i$ where $I_i$ is the $i^{th}$ measurement and $<I>$ is the weighted mean of all measurements of $I$.

[b]$R_{pim} = \Sigma_{hkl}1/(N-1)^{\frac{1}{2}} \Sigma_i|I_i(hkl) - \overline{I(hkl)}|/\Sigma_i I(hkl)$, where N is the redundancy of the data and $I(hkl)$ the average intensity.

[c]$<I /\sigma I>$ indicates the average of the intensity divided by its standard deviation.

[d]$R_{work} = \Sigma_{hkl}||F_o| - |F_c|| / \Sigma_{hkl}|F_o|$ where $F_o$ and $F_c$ are the observed and calculated structure factor amplitudes.

[e]$R_{free}$ same as R for 5% of the data randomly omitted from the refinement. The number of reflections includes the $R_{free}$ subset.

[f]Ramachandran statistics were calculated with MolProbity in PHENIX.

Numbers in parentheses refer to the respective highest resolution data shell in the dataset.

(Supplementary Fig.1b) and, due to fewer missing residues, the A chain will be discussed here.

Like UBA1 as well as the NEDD8 E1 and SUMO E1 enzymes, UBA6 belongs to the canonical class of E1 enzymes[21] and hence shares their conserved multidomain architecture (Supplementary Fig. 1c–e) consisting of a core composed of the pseudo-symmetrically arranged AAD and IAD domains together with a helical-bundle domain. The latter entity was traditionally assigned as a 4-helix bundle, however, as one of the helices rather belongs to the IAD we will refer to this as the 3-helix bundle (3HB). This compact core is decorated with the FCCH and SCCH domains on one side and the UFD on the other, resulting in a Y-shaped molecule (Fig. 1a–c). The cores of the two UBA6 molecules present in the asymmetric unit exhibit almost identical conformations with a root mean square deviation (rmsd) of 0.26 Å (Supplementary Fig. 1c). Taking the average coordinate error into account, as reflected in the data precision index with 0.39 Å, this part of the structure is virtually invariant. However, the accessory components, the FCCH and SCCH domains as well as the UFD, display enhanced variability around this conserved core. The UFD, in particular, adopts slightly different orientations as already observed and described for Uba1[22,23]. Individually, these domains can be superimposed with comparable rmsd values as the core, however, they undergo rigid body motions. Con-

sequently, a superimposition of the two full-length UBA6 molecules results in an rmsd of 2.13 Å for 986 (out of a total of 1017) aligned Cα atoms. This larger rmsd is due to the aforementioned motions of the FCCH, SCCH and UFD domains, which are linked to the core via flexible loops (Supplementary Fig. 1c).

The FCCH domain is tethered to the IAD and 3HB domains by the β7 and β15 loops, respectively. Likewise, the SCCH domain is linked to the AAD by two loops which are generally referred to as crossover and reentry loops (Fig. 1a–c), with the first leading into the SCCH domain and the latter connecting this domain back to the core. Finally, the UFD is tethered to the AAD by a single loop. The crossover loop traverses from one side of the molecule to the other and was reported to play an important role in directing the ubiquitin/UBL C-terminal tail into the adenylation site in the respective E1[22]. The UFD and SCCH domains are positioned across from each other forming a large canyon in between (Fig. 1a, b) that serves to accommodate the E2 enzymes during the transthioesterification reaction that transfers ubiquitin/UBL from the active site cysteine residue in the E1 to its counterpart in the E2. The multidomain structure and plasticity of the canonical E1s were reported to (1) differentiate the UBLs for activation which involves many specific interactions and requires large conformational changes[24,25] and (2) selectively recruit the E2s and subsequently transfer the UBLs

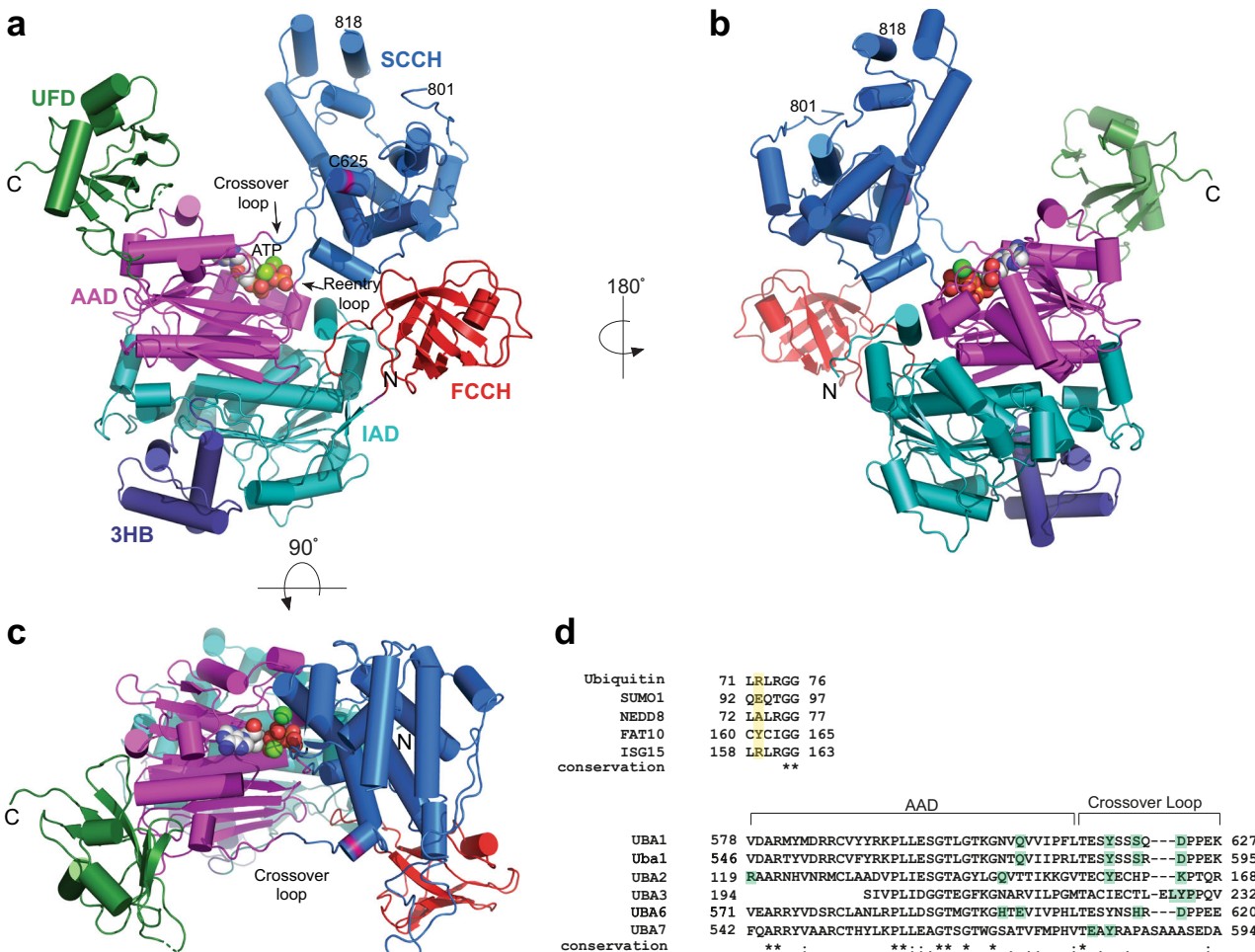

**Fig. 1 | Overall structure of human UBA6. a** Front, (**b**) back and (**c**) top views of the UBA6-ATP complex with the inactive adenylation domain (IAD) in cyan, the active adenylation domain (AAD) in magenta, the first catalytic cysteine (FCCH) domain in red, the second catalytic cysteine (SCCH) domain in marine blue, the 3-helix-bundle (3HB) in dark blue and the ubiquitin fold domain (UFD) in dark green. The crossover and reentry loops are indicated and the position of the active site C625 is highlighted in pink. ATP and coordinating metal ions (green) are shown in CPK representation. **d** Sequence alignment of the last six residues of ubiquitin and selected Ubls with the specificity determinant highlighted in yellow (top). Partial sequence alignment of the AAD and crossover loop in canonical E1 enzymes (bottom). Residues known (Uba1, UBA1, UBA2, UBA3) or predicted (UBA6, UBA7) to interact with the specificity determinant are highlighted in green.

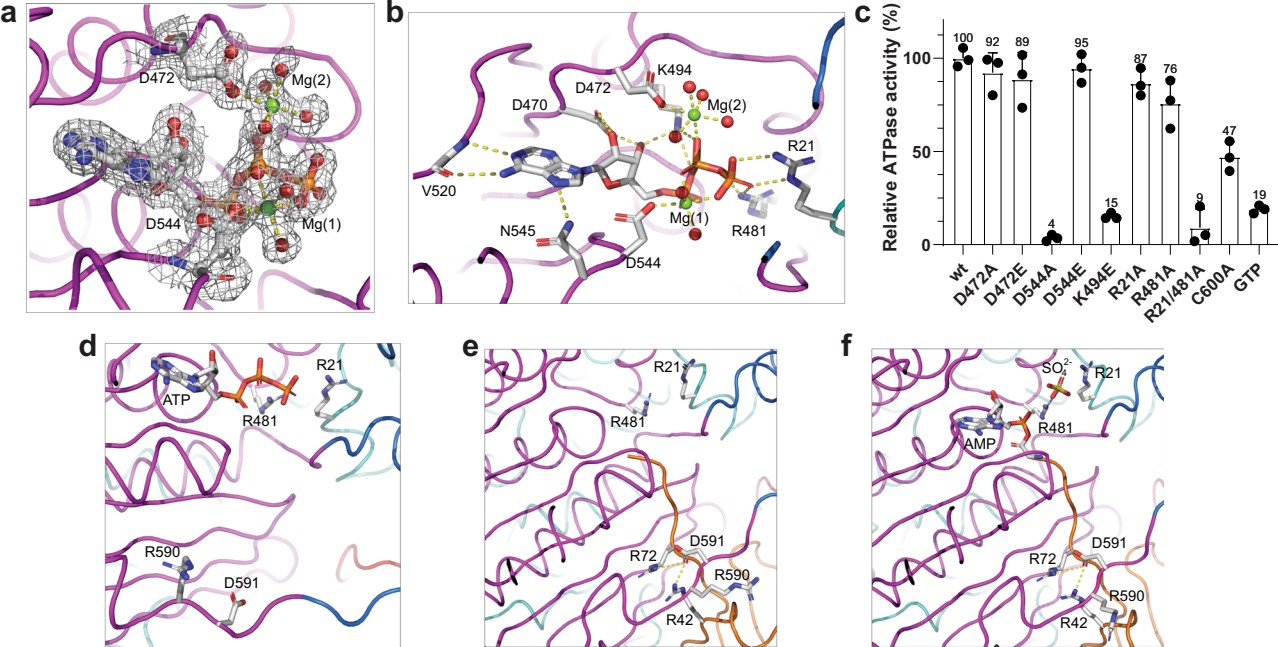

**Fig. 2 | Comparison of the Mg²⁺-ATP-UBA6 and Mg²⁺-ATP-Uba1 complexes.**
**a** SIGMAA-weighted $2F_o$-$F_c$ electron density for ATP, the two Mg²⁺-ions (Mg(1) and Mg(2)) and the coordinating Asp residues in the high resolution Uba1-ATP complex. **b** Coordination of the nucleotide cofactor in the high resolution Uba1-ATP structure with direct interactions highlighted as dashed yellow lines. **c** ATPase activity assays of Uba1 wild-type and variants, $n = 3$ independent replicates, data are presented as mean values ± SD. Source data are provided as a Source Data file. Comparison of the nucleotide binding sites and crossover loops in the (**d**) Uba1-ATP complex (this study), (**e**) the Uba1-Ub complex (PDB entry 6zqh) and **f** the Uba1-Ub-AMP complex (PDB entry 4nnj). Key residues are displayed with their side chains and are labeled.

from the E1s to the E2s²⁵,²⁶. With respect to UBL differentiation, the residue corresponding to R72 of ubiquitin is the primary specificity determinant and hence is of special importance for its activation by UBA1. The corresponding residues in NEDD8, SUMO1–3 and FAT10 are Ala, Gln/Glu (SUMO1,2/3) and Tyr, respectively (Fig. 1d).

## ATP binding and hydrolysis

As the resolution of the UBA6-ATP complex did not permit a detailed analysis of the interactions between Mg²⁺-ATP and the protein, in particular the coordination of the two metal cations, which could be visualized in the maps, we determined the structure of yeast Uba1 in complex with Mg²⁺-ATP at high resolution. These crystals diffracted X-rays also anisotropically, and, after processing with Staraniso, resolutions of 1.72 Å in the best direction and 2.24 Å in the weakest direction were obtained. Refinement resulted in a high-quality model (Table 1 and Supplementary Fig. 2a, b) in which the majority of residues, including the bound ATP together with bound metal ions and the surrounding residues, were well defined (Fig. 2a). The weakest density was seen for the C-terminal UFD where residues 969–975 could not be modeled and, to a small extent, the distal region of the SCCH (Supplementary Fig. 2a). At present, this represents the highest resolution structure obtained for a canonical E1 enzyme.

ATP is very well defined in the electron density map (Fig. 2a) and is bound in the nucleotide binding pocket located in the AAD domain as first observed in the structure of the MoeB-MoaD complex²⁷. This study revealed that the homodimeric MoeB features two Mg-ATP binding sites, with the majority of residues being contributed by one protomer, however, one critical arginine located close to the N-terminus also contacted the triphosphate moiety. In canonical E1 enzymes only one active site remains in the AAD, however, the aforementioned N-terminal arginine resides in the IAD. Somewhat surprisingly, two octahedrally coordinated magnesium ions instead of one were observed to interact with the triphosphate moiety of the nucleotide (Fig. 2b). The first Mg-ion, Mg(1), is coordinated by the side chain of D544, one oxygen each of the α, β and γ phosphates and two water molecules, while Mg(2) interacts with the side chain of D472, one oxygen of the β phosphate and four water molecules. The triphosphate moiety is also recognized by R21 from the IAD and R481 from the AAD which interact with the γ phosphate and the oxygen atom bridging the β and γ phosphate. Finally, K494 contacts the bridging oxygen between the α and β phosphate and the 3′ OH-group. The 2′ and 3′ OH-groups of the ribose engage in H-bonds with the carboxylate of D470, while the base interacts with the side chain amide of N545 and both main chain atoms of V520. The interactions between ATP and protein are completely conserved in the UBA6-ATP complex (Supplementary Fig. 2c), thus illustrating the common framework for ATP-binding and hydrolysis in the E1 enzyme family. Due to the limited resolution of the UBA6-ATP complex the metal coordination cannot be fully resolved, however, a Mg²⁺-ion was modeled at the Mg(1) site, while a Ca²⁺-ion appears to be present in the Mg(2) site (Supplementary Fig. 2c).

The function of the coordinating residues was probed by site directed mutagenesis and a colorimetric ATPase activity assay (Fig. 2c). Comparing the contribution of the two acidic residues revealed that only D544 is critical for ATP hydrolysis since the D544A variant was almost completely impaired (Fig. 2c). In contrast, the D472A variant retained full activity as did the D472E or D544E substitutions. This indicates that the Mg(1) ion is the catalytically critical cation and that Mg(2) may not be present under physiological Mg-concentrations. While the individual R21A or R481A variants only marginally impaired the ATPase activity, the corresponding double mutant drastically lowered the activity. The reverse charge mutation of K494E also dramatically affected catalysis. Interestingly, substitution of the catalytic cysteine, which is located far away from the adenylation site in the open state observed for both the Uba1-ATP and UBA6-ATP complexes, also diminished the catalytic activity by half. Since the release of the ubiquitin adenylate requires the catalytic cysteine, this is reasonable, as a non-hydrolysable ubiquitin adenylate can act as potent inhibitor for Uba1 with a $K_d$ value in the low pM range²⁸. Finally, this assay

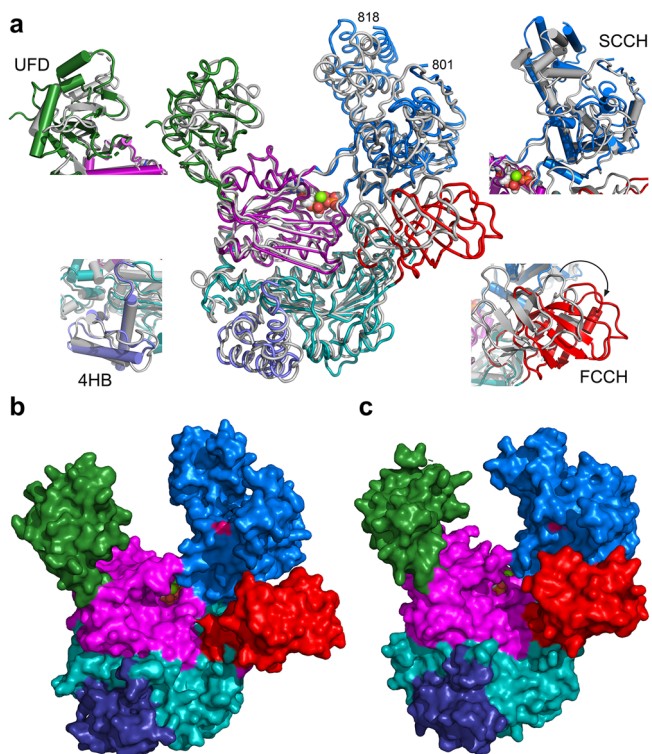

**Fig. 3 | Comparison of the UBA6 and Uba1 structures. a** Superimposition of the UBA6-ATP complex color coded by domains as in Fig. 1 with the Uba1-ATP complex in gray with both structures in loop representation. The Mg-ATP cofactor is shown in space filling representation. Differences in the 4HB, FCCH, SCCH and UFD are highlighted in ribbon diagrams utilizing the same color code. **b** Surface representation of the UBA6-ATP complex color coded by domains with the active site cysteine highlighted in pink. **c** Surface representation of the Uba1-ATP complex.

confirmed that GTP is not a suitable substrate for Uba1, which appears to be due to the incompatible arrangement of the main chain oxygen of V520 and the guanosine exocyclic keto-group, i.e. two hydrogen bond acceptors and not because of steric repulsion of the exocyclic amino group as this exchange seems to be tolerated well and, in fact, appears to be stabilized by hydrogen bonds (Supplementary Fig. 2d).

**Comparison of Uba1 in complex with either ubiquitin or ATP**
Together with the Uba1-Ub complex at 2 Å resolution (PDB entry 6zqh), the Uba1-ubiquitin-acyladenylate complex (PDB entry 4nnj) at 2.4 Å resolution and the Uba1-ATP complex (this study) three structures of Uba1 are now available, which are highly relevant for the adenylation reaction (Fig. 2d–f). A comparison of these structures revealed very high structural conservation in the ATP-binding pocket. Irrespective of the presence of either ATP or the ubiquitin-AMP adenylate the critical residues R21 and R481 adopt identical conformations, which also applies to D544 (not shown in Fig. 2d–f), thus indicating that the ATP-binding pocket is preformed and ready to accept Mg-ATP. In contrast, significant differences can be observed in the crossover loop with the Uba1-Ub and Uba1-Ub-AMP complexes adopting closely related conformations while that of the Uba1-ATP complex differs. This change can be readily visualized by inspecting the conformation of the R590 and D591 side chains. In the Uba1-ATP complex the side chain of R590 points, to a first approximation, towards the bound ATP while it is reoriented away from the nucleotide binding site in both ubiquitin-bound structures. Instead, R42 of ubiquitin displaces R590 of Uba1 and engages in ionic interactions with D591, which in the ubiquitin-bound structures is reoriented, contacting not only R42 but also R72 of ubiquitin (Fig. 2e). The role of R72 as ubiquitin specificity determinant has been described in detail earlier[22].

however, in the absence of the ATP-bound state the ubiquitin-induced structural changes could not be analyzed.

**Comparison of the UBA6 and Uba1 structures**
The structure of the Uba1-ATP complex provides an excellent framework to compare it to the UBA6-ATP complex as they both represent the same step in the activation cascade. Although the structure of human UBA1 in complex with ATP is available[23], we focused on the yeast Uba1-ATP complex due to its higher resolution and the fact that only the triphosphate moiety of ATP could be visualized in the human UBA1-ATP complex, nevertheless, the structural differences described below also apply to UBA1. Fig. 3a displays the results of the corresponding superimposition, which reveals a maximal agreement for the core region composed of the AAD-IAD-3HB domains and larger deviations in the peripheral domains. In particular, the FCCH domain in UBA6 exhibits a movement of 7.5 Å away from the AAD-IAD core and hence away from where the ubiquitin molecule binds in the AAD domain as well as a 29° rotation. This change consequently introduces a broader cleft between the FCCH domain and the core of UBA6 (Fig. 3b, c). At the same time, interactions between the FCCH and SCCH are formed. In the B-chain (Supplementary Fig. 3a) there are extensive contacts involving a hydrophobic core formed by F217, L232 and I287 of the FCCH domain and the aliphatic part of S737 as well as P738 of the SCCH domain and two ion pairs involving E249 (FCCH) and R740 (SCCH) as well as R248 (FCCH) and E876 (SCCH). Due to an even more pronounced FCCH-movement in chain A (Supplementary Fig. 1c) only the ion pair involving R248 (FCCH) and E876 (SCCH) stabilizes the two domains (Supplementary Fig. 3b). In contrast, there is a conserved interaction of the residues corresponding to E249 and R740 in UBA1 orthologs (Supplementary Fig. 3c), which is only possible with the FCCH positioned close to the IAD. One attractive hypothesis derived from the outward movement and rotation of the FCCH would predict that the extra space in between the FCCH and IAD is required to accommodate the second ubiquitin-like domain of FAT10, at least at one point during the catalytic cycle.

**Overall Structure of the UBA6-FAT10 complex**
While predictions regarding the orientation of ubiquitin in complex with UBA6 can be readily generated on the basis of the available Uba1-Ub or Uba1-Ub-AMP structures, it seemed impossible to predict how FAT10 and, in particular, its N-terminal domain (NTD) would interact with UBA6. One hypothesis would posit a placement of the NTD in the large cleft between the AAD and FCCH present in the UBA6 structure. Hence, we generated the UBA6-FAT10 complex and solved its structure by molecular replacement with full-length UBA6 as search model. Due to superior stability and solubility a FAT10 variant described earlier[11] in which Cys7 and Cys9 were substituted with Thr, Cys134 was replaced with Leu and Cys160 and Cys162 were replaced with Ser (referred to as FAT10-C0) was generated. Since extensive initial crystallization attempts with wild-type UBA6 did not result in well diffracting crystals, a chimeric protein, referred to as UBA6$_{chim}$, was utilized. In this protein the SCCH domain (residue 623 to 889) of UBA6 was replaced with its counterpart (residue 631 to 889) from human UBA1, resulting in a protein with enhanced stability. Activity assays with FAT10-C0 showed a robust activation by UBA6 and transthioesterification to USE1 (Supplementary Fig. 4a, b), which was significantly stronger than that of the FAT10 wildtype, possibly due to the increased solubility and/or stability of FAT10-C0. In the activity assay with FAT10-C0 two additional bands above the UBA6 band are visible with the more slowly migrating band showing weaker intensity. We attribute the lower band to the thioester-linked UBA6-FAT10-C0 adduct and the upper band to a UBA6 protein which also contains FAT10-C0 linked via an isopeptide bond. DTT treatment is known to disrupt the thioester linkage and, correspondingly, we observe only one band after DTT treatment which runs at the height of the lower band, however, with an

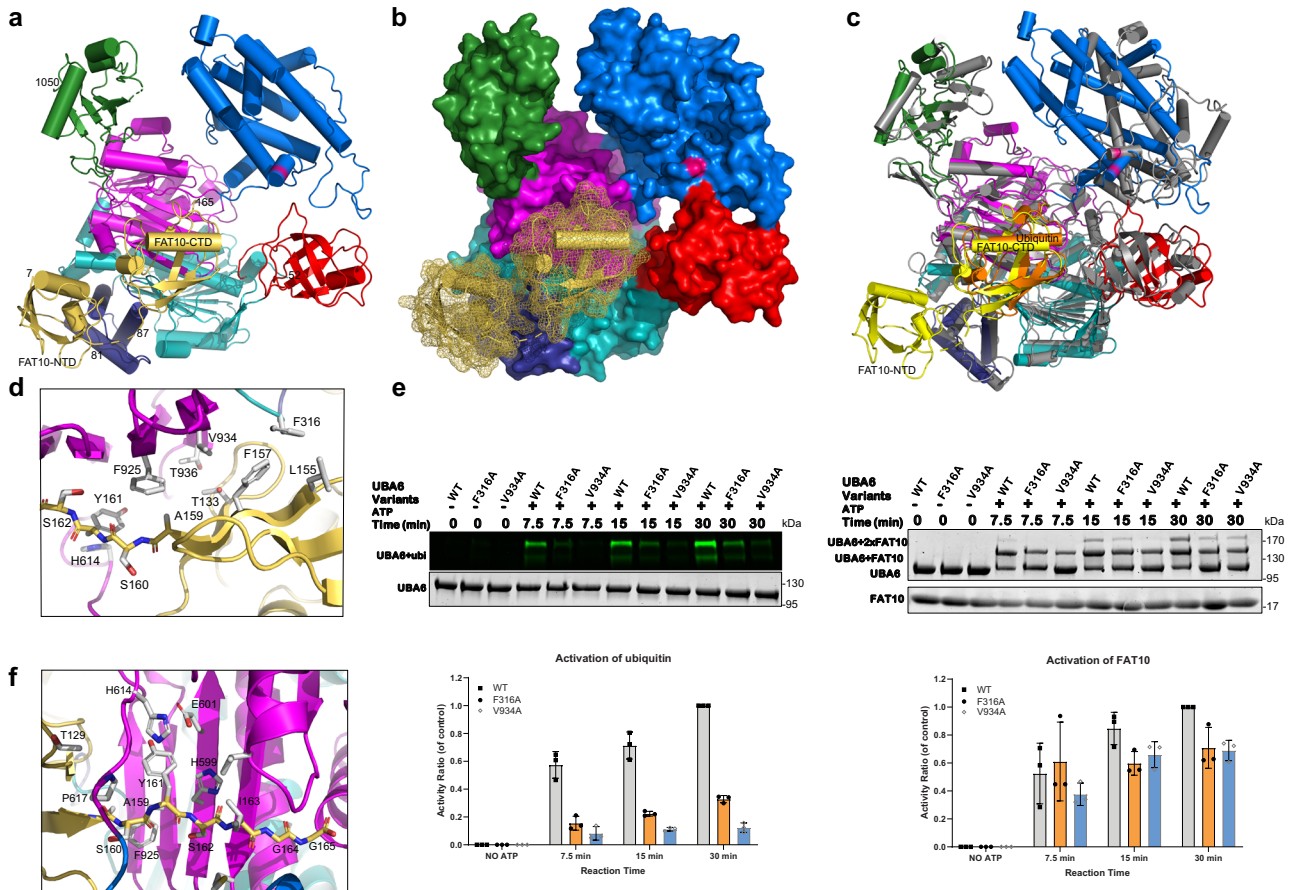

**Fig. 4 | Structure of the UBA6-FAT10 complex. a** Overall structure of UBA6 color coded according to its domains in complex with FAT10 in yellow. The NTD and CTD of FAT10 are labeled. **b** UBA6-FAT10 complex with UBA6 in surface representation and FAT10 in ribbon and dot representation. **c** Superimposition of the UBA6-FAT10 complex color coded as in (**a**) and the Uba1-ubiquitin complex with Uba1 in gray and ubiquitin in orange. **d** Hydrophobic interface between the CTD of FAT10 (yellow) and AAD (magenta)/IAD (cyan) of UBA6. Interacting residues are

labeled and shown with their side chains. Parts of the FAT10 C-terminal tail (residues 159–162) are shown in all-atom representation. **e** Ubiquitylation (left) and FAT10ylation (right) assays of UBA6 wild-type and variants (F316A and V934A). Raw data (top) and quantification (bottom). $n = 3$ independent replicates, data are presented as mean values ± SD. Source data are provided as a Source Data file. **f** Recognition of the FAT10 C-terminal tail in all-atom representation with surrounding residues in the AAD and IAD.

intensity corresponding to that of the upper band in the sample not treated with DTT (Supplementary Fig. 4c).

The P1 crystals contain two copies of UBA6$_{chim}$, which could be positioned by molecular replacement using a search model from which the SCCH domain was omitted. The two SCCH domains were localized subsequently by molecular replacement with the corresponding domain of UBA1 (PDB entry 6dc6), followed by extensive rebuilding. Surprisingly, only a single FAT10 moiety was present, which was assembled from the crystal structure (PDB entry 6gf1) of its NTD[11] and the NMR structure of its C-terminal domain (CTD, PDB entry 6gf2) with the loop connecting the two domains apparently exhibiting high flexibility. The overall structure of the UBA6$_{chim}$-FAT10 C0 complex revealed that FAT10 covers the AAD, IAD and 3HB domains of UBA6$_{chim}$ and engages in numerous interactions distributed over a smaller and a larger interface involving the NTD and CTD, respectively (Fig. 4a, b). The UBA6-FAT10 interface encompasses an area of 1725 Å$^2$, which corresponds to 17.7% of the FAT10 total surface area. Breaking this down into the contributions by the NTD and CTD, respective interface areas of 478 Å$^2$ and 1259 Å$^2$ were calculated. In comparison, in the Uba1-ubiquitin and UBA1-ubiquitin complexes (PDB entries 6zqh and 6dc6) the interface areas amount to 1621 Å$^2$ and 1561 Å$^2$, respectively, which clearly exceeds the value for the CTD in the UBA6-FAT10 interface.

A superimposition of the Uba1-ubiquitin and the UBA6$_{chim}$-FAT10 complexes (Fig. 4c) revealed high structural conservation for the IAD, 3HB, AAD and UFD domains. Minor changes were observed for the FCCH, which in the UBA6$_{chim}$-FAT10 complex is not displaced outward as much as in the Uba1-ubiquitin complex. While the SCCH domains of UBA6$_{chim}$ and its counterpart in UBA1 and Uba1 adopt quite similar structures in isolation (Supplementary Fig. 4d), the SCCH in UBA6$_{chim}$ undergoes drastic conformational changes. Compared to the Uba1-ubiquitin complex the SCCH is rotated by more than 60° towards the UFD coupled to a displacement of the crossover loop by up to 10 Å near the active site cysteine. These conformational changes are observed in both copies of UBA6$_{chim}$ which exhibit almost identical structures (Supplementary Fig. 4e) and hence do not seem to be induced by FAT10 binding and may be due to the chimeric nature of this construct. In contrast, significant conformational changes between the UBA6$_{chim}$-FAT10 complex and the second UBA6$_{chim}$ molecule (Supplementary Fig. 4f) were observed only for the N-terminal region of the crossover loop involving residues 614–619, including R615 and D616. These changes are in analogy to what was observed for R590 and D591 of Uba1 (Fig. 2d–f), where binding of ubiquitin also leads to changes in the crossover loop. Apparently, these residues act as sensors for the presence of Ub/Ubl bound at the adenylation site.

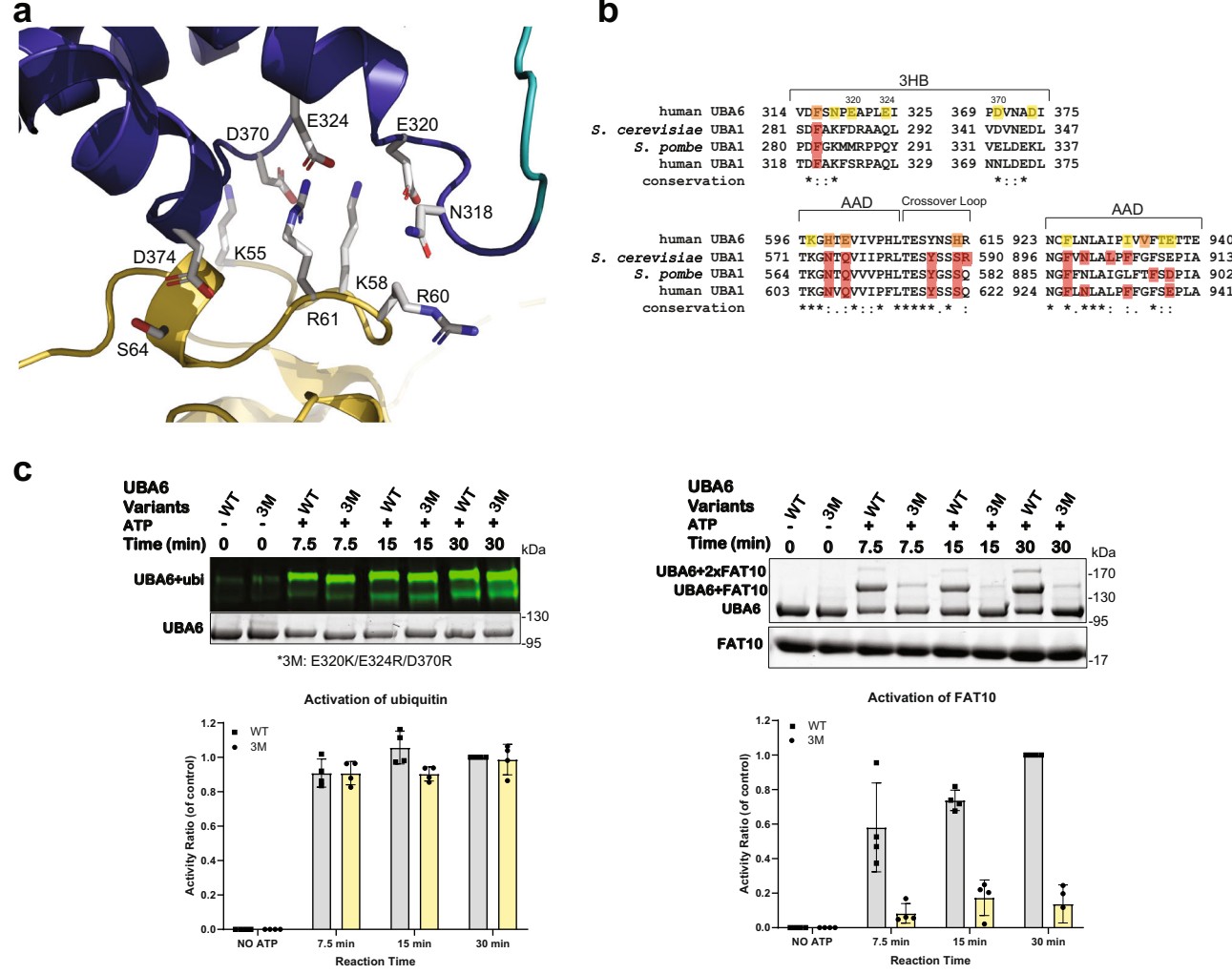

**Fig. 5 | UBA6-FAT10(NTD) interface and selective elimination of FAT10ylation.**
**a** Interactions between the FAT10 NTD (yellow) and the 3HB (dark blue) of UBA6.
**b** Partial sequence alignment of UBA6 and UBA1 orthologs highlighting residues
only interacting with either FAT10 (yellow) or with ubiquitin (red) and residues
which interact with both FAT10 and ubiquitin (orange). **c** Ubiquitylation (left) and

FAT10ylation (right) assays probing the contributions of the negatively charged
residues (triple mutant E320K/E324R/D370R, referred to as 3 M) in the 3HB of
UBA6 in UBA6-FAT10 complex formation. Raw data (top) and quantification
(bottom). $n = 4$ independent replicates, data are presented as mean values ± SD.
Source data are provided as a Source Data file.

## Analysis of the UBA6-FAT10 Interfaces

The CTD of FAT10 is bound to UBA6$_{chim}$ in an analogous manner as
ubiquitin binds to Uba1 and the interactions can be grouped primarily
into two areas:

First, multiple hydrophobic interactions are formed between the
CTD and mostly the AAD with minor contributions from the IAD. In the
Uba1-ubiquitin complex these interactions involve the classical
hydrophobic patch including Ile44 of ubiquitin, however, this residue
is not conserved in the FAT10 CTD where it is replaced by T133
(Fig. 4d). While T133 engages in limited van der Waals interactions
involving its side chain Cβ and Cγ atoms, more prominent hydro-
phobic contacts are formed by L155, F157 and A159 of FAT10 with F316
in the IAD as well as F925, V934 and, to a more limited extent, T936,
which are both located in the AAD.

UBA6 variants were engineered to probe the interactions between
the AAD and either ubiquitin or the FAT10 CTD (Fig. 4e). The removal
of the side chain in the F316A variant reduced the activation of ubi-
quitin significantly (33%) and, to a more limited degree, that of FAT10
(71%). Likewise, the V934A variant strongly reduced the activation of
ubiquitin (12%) and mildly impaired FAT10 activation (69%). These data
confirmed that FAT10 and ubiquitin bind to the same hydrophobic
platform on the AAD with minor contributions from the IAD and

weakening these interactions resulted in a reduction of UBA6 with
respect to both modifiers. The smaller reduction in FAT10 activation
may be attributed to the fact that the NTD of FAT10 is still capable of
binding to UBA6, thus retaining higher activity levels (Fig. 4e).

Second, the C-terminal tail of FAT10 is threaded underneath the
crossover loop towards the ATP binding site with the C-terminal G165
being in close proximity to where the α-phosphate resides in the UBA6-
ATP complex (Fig. 4f). With the exception of the C-terminal Gly-Gly-
dipeptide, the last six residues of FAT10 ($^{160}$Cys-Tyr-Cys-Ile-Gly-Gly$^{165}$)
differ substantially from the corresponding residues of ubiquitin
($^{71}$Leu-Arg-Leu-Arg-Gly-Gly$^{76}$), and a key question regarding the dual
specificity of UBA6 is how the respective specificity determinants R72
and Y161 can both be recognized by UBA6. The UBA6$_{chim}$-
FAT10 structure clearly revealed that the same binding pocket used by
Uba1 to recognize R72 of ubiquitin is recruited for the binding of Y161
of FAT10 (Fig. 4f). In contrast to the polar interactions involving the
recognition of R72, Y161 is recognized predominantly by hydrophobic
interactions involving H599 and H614, which are replaced by Asn and
Ser (N574 and S589 in Uba1; N606 and S621 in UBA1). The presumably
protonated side chain of H614 appears to form a cation-π interaction
with the aromatic ring system of Y161 and is stabilized in turn by an
interaction with E601 (Fig. 4f). The residue in UBA1/Uba1

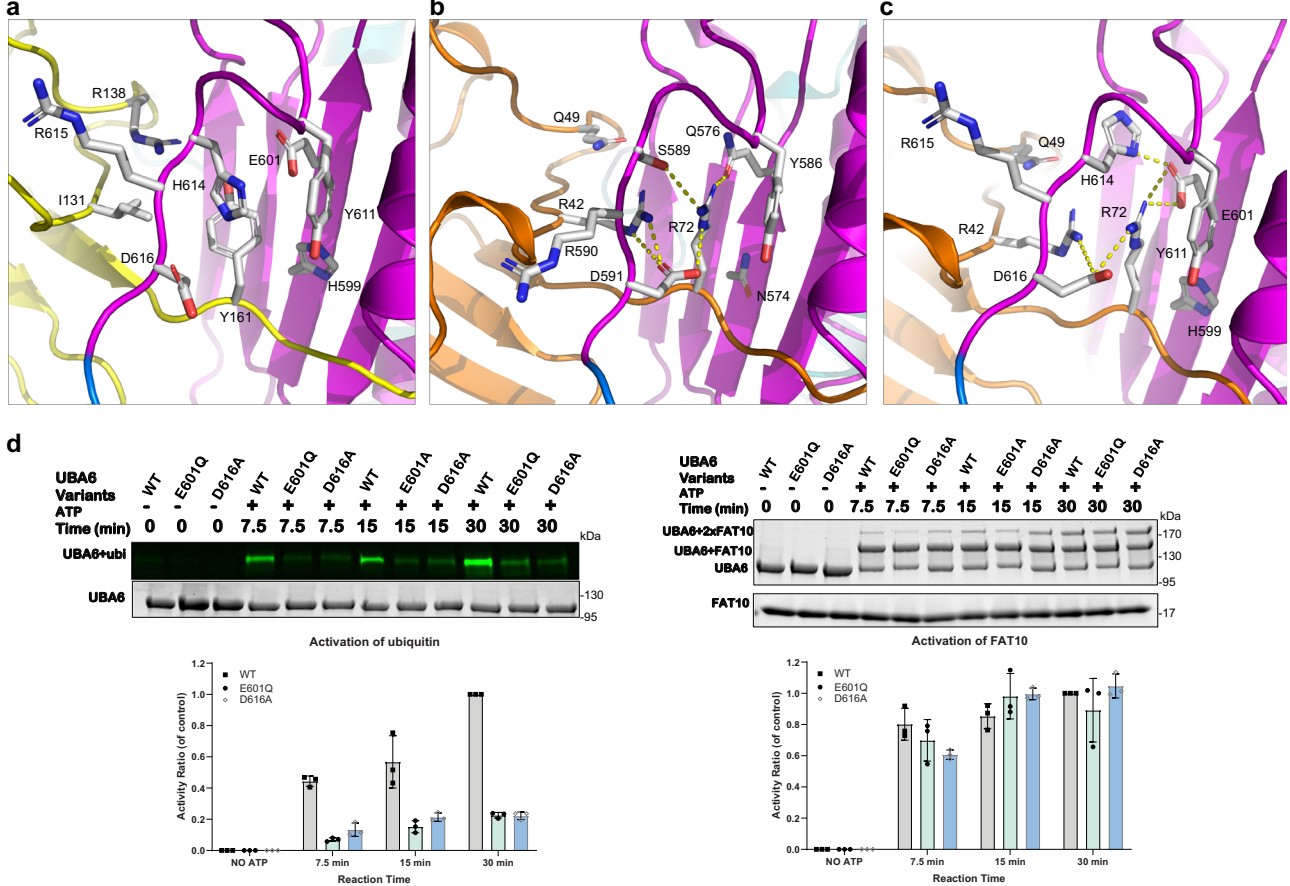

**Fig. 6 | Molecular basis of the dual specificity of UBA6 and selective abrogation of ubiquitylation. a** Recognition of the Y161 specificity determinant of FAT10 (yellow) by UBA6 in a predominantly hydrophobic pocket. **b** Interactions between R72 of ubiquitin (orange) and Uba1 (PDB entry 6zqh) with polar interactions indicated as dashed yellow lines. **c** Model of the UBA6-ubiquitin complex.

Compared to (**a**) H614 and D616 adopt different side chain rotamers. **d** Ubiquitylation (left) and FAT10ylation (right) assays of the UBA6 D616A and E601Q variants. Raw data (top) and quantification (bottom). $n = 3$ independent replicates, data are presented as mean values ± SD. Source data are provided as a Source Data file.

corresponding to E601 is a Gln (Q608/Q576 in UBA1/Uba1) and is a key element for the recognition of ubiquitin's R72 by forming a hydrogen bond involving its side chain oxygen and one of the terminal N-atoms of R72[22].

In contrast to the hypothesis that the NTD of FAT10 is placed in between the AAD and FCCH and interacts with these domains (see "Comparison of the UBA6 and Uba1 structures"), it is oriented in such a way that it exclusively interacts with the 3HB inserted into the IAD. These interactions are polar and primarily rely on the electrostatic complementarity of a positively charged region in the NTD involving residues K55, K58, R60 and K61 and a negatively charged patch in the 3HB formed by E320, E324, D370 and D374, with S64 of FAT10 and N318 of UBA6 contributing additional polar interactions (Fig. 5a, b, Supplementary Fig. 4g). At the same time, the NTD also contacts the second UBA6 molecule (B-chain) in an interaction which mimics binding of the CTD to the A-chain (Supplementary Fig. 5a–e). Given the sequence identity levels between ubiquitin and FAT10 of 29% and 36% for the NTD and CTD, respectively, it is not entirely surprising that the NTD can occupy the position of the CTD in its interaction with the B-chain of UBA6, although the level of sequence identity between the NTD and CTD of FAT10 is only 18% (Supplementary Fig. 5e). Nevertheless, this raised the possibility that the observed binding mode could be a crystallization artefact, driven mostly by the interaction of the NTD with the B-chain. We therefore probed the contribution of residues in the NTD-3HB interface by site directed mutagenesis. Substitution of the negatively charged residues E320, E324 and D370 in

UBA6 with oppositely charged residues (E320K, E324R and D370R) significantly impaired (Fig. 5c) the activation of FAT10 to (14%) by UBA6, while still retaining full activity towards ubiquitin (99%). These studies hence confirmed that the interaction observed between the NTD of FAT10 and the 3HB of UBA6 has functional significance and does not represent a crystallization artifact. More importantly, the UBA6 E320K, E324R and D370R triple variant represents a suitable tool to study a UBA6 variant deficient in FAT10ylation when engineered into UBA6 knockout cells.

**Modeling the UBA6-ubiquitin interaction and probing the predicted complex**

After superimposing UBA6 and Uba1 in complex with ubiquitin and comparing in detail the interactions involving the selectivity determinants Y161 of FAT10 in the UBA6-FAT10 complex (Fig. 6a) and R72 of ubiquitin in the Uba1-ubiquitin complex (Fig. 6b) allowed us to generate a model of the UBA6-ubiquitin complex (Fig. 6c, Supplementary Fig. 6a). As mentioned above, the environment of Y161 is substantially more hydrophobic than that of R72, which is due to the replacement of S589 in Uba1 (S621 in UBA1) with H614 in UBA6 and the substitution of I131 in FAT10 for R42 in ubiquitin. At the same time, D616 of UBA6 in the UBA6-FAT10 adopts a different side chain conformation compared to its counterpart D591 in Uba1 (D623 in UBA1) and is pointing away from Y161 towards the solvent. In the model of the UBA6-ubiquitin complex, a favorable environment for R72 can be easily created by modeling D616 with a side chain conformation closely mimicking its

counterpart D591 in Uba1 while, at the same time, the side chain of H614 is reoriented so that it can form polar interactions with E601 of UBA6 (Supplementary Fig. 6b).

This analysis predicts that elimination of the negatively charged D616 should weaken the interaction between UBA6 and ubiquitin, while showing no effect for UBA6-catalyzed FAT10ylation. Hence, we engineered the D616A variant and analyzed its activity with both protein modifiers. While FAT10ylation levels for the variant remained at wild-type levels (105% of wild-type), ubiquitylation was severely impaired resulting in a residual activity of 22% for the mutant (Fig. 6d). D591, the corresponding residue in Uba1, undergoes conformational changes upon ubiquitin binding as described earlier (Fig. 2d–f). These changes, together with the critical role of D616 in the UBA6-catalyzed ubiquitylation, support the notion that this aspartate, in conjunction with the preceding arginine, not only acts as a sensor detecting the presence of ubiquitin in Uba1 but also as a molecular switch shifting the preference of UBA6 towards either ubiquitin or FAT10.

The counterpart of E601 in Uba1 is Q576 (Q608 in UBA1), which, albeit being polar, cannot contribute towards the neutralization of the positively charged R72 in the same way as predicted for E601 (Fig. 1d). Hence, the E601Q variant of UBA6 was generated and analyzed in activity assays (Fig. 6d). While the activation of FAT10 was hardly affected by this substitution, retaining 89% of the wild-type activity, a strong reduction in ubiquitin activation was observed resulting in a residual activity of 22% (Fig. 6d). Thus, our model of the UBA6-ubiquitin complex (Fig. 6c), would predict that the E601Q variant disrupts the binding pocket for R72 since the amide in the side chain of the Gln would clash either with R72 of ubiquitin or H614 of UBA6. Apparently, H614 is not able to adopt an alternative favorable side chain conformation, presumably due to the presence of R42 and R72 of ubiquitin. Thus, the D616A and E601Q variants selectively impair ubiquitylation while retaining the FAT10ylation activity in UBA6 and hence are excellent tools to study the cellular effects of UBA6-mediated ubiquitylation.

## Discussion

The structures of UBA1, NEDD8 E1 and SUMO E1 provided significant insights into the mechanism of how canonical E1s activate their cognate Ubls. In this study, we report the structure of UBA6 which is the fourth member of the canonical E1 family to be structurally characterized. This leaves UBA7, the activating enzyme for ISG15, which, like FAT10, consists of tandem Ubl domains, as the sole member for which currently no high-resolution structural data are available. A superimposition of the ISG15 structure[29] (PDB entry 1z2m) reveals that the relative orientation of the two Ubl domains in both of these modifiers is similar (Supplementary Fig. 7a), thus suggesting that the NTD of ISG15 also interacts with the 3HB of UBA7 (Supplementary Fig. 7b). The UBA7 Alphafold-predicted model (https://alphafold.ebi.ac.uk/entry/P41226) retains the outward movement of the FCCH observed for UBA6 (Fig. 3a, b), thus supporting the notion that this is a conserved feature of canonical E1 enzymes recognizing tandem-repeat Ubl-modifiers, i.e. FAT10 and ISG15. Since the extra space between the AAD and FCCH is not required for recruiting FAT10 and, by analogy, ISG15 to the adenylation active site, one hypothesis would be that this extra space is necessary to support the NTD of either modifier when bound as a thioester to the active site cysteine. Modeling studies (data not shown) support this hypothesis.

The involvement of UBA6-mediated ubiquitylation and FAT10ylation in a broad spectrum of cellular processes and diseases renders UBA6 an attractive drug target, however, the dual specificity of UBA6 complicates this issue. Understanding and deciphering this dual specificity hence represents a prerequisite for the targeted inhibition of UBA6, ultimately leading to the inhibition of FAT10ylation while not affecting UBA6-mediated ubiquitylation and vice versa. A key aspect of this study is therefore the identification of UBA6 variants which

selectively abolish activation of either modifier. Specifically, by modeling the UBA6-ubiquitin complex and comparison with the Uba1-ubiquitin structure, E601 of UBA6 was identified which, after substitution with a Gln, severely impaired the activation of ubiquitin (Fig. 6d) while, at the same time, not interfering with the activation of FAT10. The same impairment of UBA6-mediatated ubiquitylation was also observed for the D616A variant. This yields two UBA6 variants which can be used to study the effects of UBA6-mediated ubiquitylation. By combining the two variants an even more potent inhibition maybe obtained. Conversely, the structure of the UBA6-FAT10 complex revealed interactions between the FAT10 NTD and the 3HB of UBA6, which are driven predominantly by electrostatic interactions. Grouped charge reversal mutations in UBA6 disrupted this interface and prevented FAT10 activation by UBA6. Consequently, the UBA6 charge reversal variant is a great tool to study the influence of UBA6-mediated FAT10ylation. These sets of mutants will now be essential tools to delineate the physiological consequences of either UBA6-catalyzed ubiquitylation or FAT10ylation. In contrast to a UBA6 knockout, which would interfere with both processes one can now selectively impair either of these processes and investigate the resulting physiological consequences. Likewise, the variants impaired in FAT10ylation are superior to a FAT10 knockout since their effects are limited to the consequences of the impaired FAT10ylation reaction, in contrast to a FAT10 knockout, which will also affect the functions of non-conjugated FAT10, which on its own serves as an important interactor of various target proteins.

## Methods
### Molecular biology
Human UBA6 (residues 35–1052) and its variants containing a glutathione S-transferase (GST) tag followed by a TEV cleavage site at the N-terminus and a His$_6$-tag at the C-terminus were cloned into the 2TK pGEX expression vector. Human UBA1 (residues 41–1058) and yeast Uba1 (residues 10–1024), which are referred to as UBA1 and Uba1, respectively, were cloned with a C-terminal His-tag in the pET23b vector and an N-terminal His-tag in the pET28a vector, respectively, and transformed into *E. coli* BL21 (DE3) RIL cells for expression. Wild-type FAT10 was N-terminally tagged with a His$_6$-SUMO tag in the pET28a vector, while a cysteine-free FAT10 (FAT10-C7T/C9T/C134L/C160S/C162S) variant, referred to as FAT10-C0, was N-terminally tagged with a His$_6$-maltose-binding-protein-tag followed by a TEV cleavage site in the pETM41 vector. The desired mutations were generated using site-directed mutagenesis with specifically designed primers (Supplementary Table 1) followed by PCR amplification. A chimeric UBA6 construct (UBA6$_{chim}$) in which the SCCH of UBA6 was replaced by its counterpart from human UBA1 was engineered using sequence and ligation independent cloning (SLIC). The respective plasmid DNAs were isolated using the Mini-prep kit (Fisher Scientific UK) and DNA sequences were verified by automated DNA sequencing.

### Protein purification
UBA6 and its variants were expressed in the *E. coli* BL21 pRARE strain and cells were grown in LB medium. Expression was induced by adding 0.2 mM isopropyl-β-D-thiogalactopyranoside (IPTG) followed by overnight growth at 15 °C. Proteins were purified by glutathione affinity chromatography, cleavage of the GST-tag with TEV protease, Ni-NTA affinity chromatography and size-exclusion chromatography. Proteins were concentrated in buffer containing 25 mM Tris-HCl pH 8, 500 mM NaCl, 1 mM TCEP and aliquoted prior to storing at −80 °C.

UBA1 and Uba1 were expressed in the *E. coli* BL21 (DE3) strain in LB medium and expression was induced by addition of 0.1 mM of IPTG and subsequent overnight growth at 16 °C. UBA1 was purified by Ni-NTA affinity chromatography, followed by anion exchange and size-exclusion chromatography. For Uba1 hydrophobic interaction chromatography replaced the anion exchange chromatography.

The FAT10 wild-type and FAT10-C0 as well as its variants were expressed in *E. coli* BL21 Rosetta II cells, respectively, in LB medium after induction with 0.2 mM IPTG followed by overnight growth at 20 °C. FAT10 proteins were purified by Ni-NTA affinity chromatography and subsequent cleavage of the $His_6$-SUMO tag with the SUMO protease ULP1 for FAT10 and cleavage with TEV protease for FAT10-C0. FAT10 proteins were further purified by cation exchange and size-exclusion chromatography. Expression and purification of USE1[30] and ubiquitin[31] were carried out as described.

## Colorimetric Uba1 ATPase assay

Eight different concentrations (0–6 μM) of Uba1 or its variants, 80 μM ubiquitin and 40 U of inorganic pyrophosphatase (SIGMA, I5907) in assay buffer (20 mM Tris, pH 7.5, 50 mM NaCl) were incubated in the presence of 1 mM ATP and 2 mM $MgCl_2$ for 30 min at 30 °C. The resulting inorganic phosphate was detected with the BIOMOL-GREEN Reagent (Enzo Life Sciences). 50 μL of each reaction was transferred into 96-NUNC microwell plates (Thermo Fisher Scientific) and 100 μL BIOMOL-GREEN reagent was pipetted to each well. After 5 min 15 μL of 34% (w/v) sodium citrate were added (Enzo life sciences, AK111-Addendum datasheet for ATP usage) followed by absorbance measurements at 620 nm using a CLARIO star plate reader (BMG Labtech, Germany). Each sample was measured in triplicates in two independent experiments. The data were normalized against a control containing 1 mM ATP, 2 mM $MgCl_2$ and 40 U pyrophosphatase in assay buffer. The data displayed a linear correlation with the concentration of Uba1 (wild-type and variants) and the specific activity was derived by a linear fit.

## E1 and E1-E2 activity assays

To test the functionalities of the purified E1s, 3 μM of the respective E1 enzyme was mixed with either 20 μM of unlabeled FAT10 wild-type, FAT10-C0 or FAT10-C0 variants (for UBA6) or 3 μM of ubiquitin (for UBA1, Uba1 and UBA6), which was labeled with the 800CW infrared fluorescent dye (IRdye, LI-COR), in the presence of 2.5 mM ATP and 2.5 mM $MgCl_2$. The reactions were conducted at room temperature for the indicated times and were stopped by adding protein loading buffer without reducing agent (50 mM Tris-HCl pH 6.8, 2% SDS, 6% glycerol, 0.01% Bromophenol blue), except when noted otherwise. SDS PAGE with 4–20% gradient gels was used for analysis. To test the transfer of FAT10 from UBA6 to USE1, 9 μM USE1 was mixed with UBA6 and Mg-ATP (as described above) and the respective FAT10 wild-type or FAT10-C0. The reaction conditions and analysis were as described for the E1 activity assay.

## Crystallization and data collection

For the UBA6/Uba1-ATP complexes either the UBA6$^{C625A}$ variant or the Uba1 wild-type was mixed in a 1:1.5 molar ratio with 1 mM ATP and 2 mM $MgCl_2$ and incubated at 4 °C overnight. In case of the UBA6-FAT10 complex the chimeric UBA6variant (UBA6$_{chim}$) was incubated with FAT10-C0 in a 1:1.5 molar ratio at 4 °C overnight. Protein crystals were obtained by vapor diffusion experiments, either in sitting drop plates or hanging drop plates at 4 °C. The UBA6$^{C625A}$-ATP complex was crystallized at a protein concentration of 4 mg/ml against a reservoir solution containing 0.12–0.16 M Ca-acetate, 0.08 M Na-cacodylate pH 6.5, 13–15% w/v PEG 8000, 16–20% v/v glycerol. The Uba1-ATP complex was crystallized at a concentration of 12 mg/ml from 0.2 M $(NH_4)_2SO_4$, 0.1 M HEPES pH 7.5 and 25% PEG 3350. The UBA6$_{chim}$-FAT10-C0 complex was crystallized at a concentration of 5 mg/ml from 0.5 M LiCl, 0.1 M Tris pH 8.4 and 25% PEG6000. All crystals were harvested in mother liquor supplement with 20–30% glycerol as cryo-protectant and flash cooled in liquid nitrogen for subsequent data collection at 100 K. Diffraction data were collected at the following synchrotron facilities: UBA6-ATP, P14 at DESY/EMBL in Hamburg; UBA6-FAT10, ID 30A-3 at ESRF-EBS in Grenoble; Uba1-ATP, BL14.1 at BESSY in Berlin.

## Structure determination and refinement

The structure of the UBA6$^{C625A}$-ATP complex was solved by molecular replacement (MR) with Phaser[32] using the yeast Uba1 structure[22] (PDB entry 3cmm) as a search model in a sequential domain by domain approach against a dataset belonging to space group $P2_122_1$ collected at a wavelength of 0.9672 Å. Based on a packing analysis two copies of Uba6 were predicted to be present in the asymmetric unit. The core of Uba1 consisting of its AAD, IAD and FCCH was used as search model for the core of UBA6 and the resulting structure was refined with Refmac[33] accounting for the substantial conformational changes in the FCCH. Subsequently, another round of MR searching for two copies of the SCCH domain was conducted and the resulting model was refined as before. Next, another round of MR was carried out to locate the two copies of the UFD, followed by refinement, which resulted in a model of the UBA6-ATP complex. After further refinement with Refmac and manual rebuilding with Coot[34] including modeling of the bound ATP, which was clearly visible in the electron density maps, phases were improved by two-fold averaging with DM. The model was extensively rebuilt and refined with Refmac and Phenix[35]. At this point the C2 dataset became available and, due to superior diffraction, refinement was continued and completed against this dataset. Anisotropy of the diffraction data was corrected with the Staraniso server (https://staraniso.globalphasing.org/cgi-bin/staraniso.cgi) from Global Phasing Limited. Data collection statistics are summarized in Table 1. The structure was solved by molecular replacement using one copy of the preliminarily refined UBA6 structure from the $P2_122_1$ dataset as search model with Phaser and refined with Phenix incorporating tight ncs restraints and TLS refinement.

The structure of the Uba1-ATP complex was solved by molecular replacement with Phaser using the structure of yeast Uba1 in complex with ubiquitin, omitting ubiquitin from the search model. The structure of the UBA6$_{chim}$-FAT10 complex was also solved by molecular replacement with Phaser using the UBA6-ATP complex as search model after the AAD had been removed. The AAD was subsequently positioned by molecular replacement followed by extensive model building. The remaining density was first interpreted by positioning the C-terminal domain of FAT10 (PDB entry 6gf2), followed by the placement of the N-terminal domain (PDB entry 6gf1). Initial refinement employed rigid body refinement to accommodate domain reorientations. Model building was conducted with Coot and refinement with Phenix. The protein structures were analyzed using The PyMOL Molecular Graphics System, Version 2.0 Schrödinger, LLC. The secondary structure depiction was generated using ESPript 3.0[36].

## Reporting Summary

Further information on research design is available in the Nature Research Reporting Summary linked to this article.

## Data availability

The following previously published PDB entries are used in this manuscript: 6dc6, 6gf1, 6gf2, 6zqh, 1z2m, 3cmm. The coordinates and structure factor amplitudes have been deposited in the PDB and can accessed as follows: UBA6-ATP: PDB entry 7pvn, Uba1-ATP: PDB entry 7zh9 and UBA6$_{chim}$-FAT10: PDB entry 7pyv. Source data for biochemical experiments are provided with this paper.

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

## Acknowledgements

The authors would like to thank the Deutsche Forschungsgemeinschaft (GRK2243; fellowships for NT and SL) and the Rudolf Virchow Center for Integrative and Translational Bioimaging for funding. M.M. was supported by a grant of the German Excellence Initiative to the Graduate School of Life Sciences, University of Würzburg. Diffraction data were collected for the UBA6-ATP complexes at beamline P14 operated by EMBL Hamburg at the PETRA III storage ring (DESY, Hamburg, Germany), for the Uba1-ATP complex at beamline 14.1 at BESSY and for the UBA6-FAT10 complex at beamline ID30A-3 of the ESRF in Grenoble, France, and we acknowledge the excellent support by the respective beamline staff. The original UBA6 construct was kindly provided by Dr. Marcus Gröttrup and the FAT10-C0 construct by Dr. Silke Wiesner. This publication was supported by the Open Access Publication Fund of the University of Würzburg.

## Author contributions

The authors made the following declaration about their contributions: (1) Uba1-ATP complex: H.S. and M.M. conceived and designed the analysis, M.M. collected the data and performed the analysis. (2) UBA6-ATP complex: H.S. and N.T. conceived and designed the analysis, N.T. collected the data and performed the analysis. (3) UBA6-FAT10 complex: H.S., N.T. and S.L. conceived and designed the analysis, S.L. collected the data, performed the experiments and analysis. H.S. and N.T wrote the paper. M.K. assisted on all wet lab experiments.

## Funding

## Competing interests

The authors declare no competing interests.
