## [Peer Review File · Nature Communications]

REVIEWERS' COMMENTS

Reviewer #1 (Remarks to the Author):

Manuscript Review for Nature Communications (May 10, 2022):

Structures of UBA6 explain its Dual Specificity for Ubiquitin and FAT10

Schindelin lab

This manuscript describes crystal structures of UBA6 in complex with ATP and with FAT10. In order to solve a structure of UBA6 with FAT10, the authors use a chimeric version of UBA6 with the SCCH domain substituted from UBA1, the related but Ubiquitin-specific E1.

First, an ATP-bound UBA6 structure is compared with Uba1-ATP to reveal the mode of ATP interaction and hydrolysis is conserved (not surprising).

More interestingly, the UBA6-FAT10 complex provides a platform for understanding the ability of UBA6 to adenylate two different ubiquitin-like proteins, ubiquitin (Ub) and FAT10.

The structure of UBA6-FAT10 reveals the C-terminal FAT10 Ubl domain is positioned in the adenylation active site (similar to ubiquitin in a Uba1 complex) while the N-terminal FAT10 Ubl domain makes additional contacts with the 3HB (three helix bundle) of UBA6. A triple point mutation within the 3HB domain reduced UBA6 charging by FAT10 but had no effect on ubiquitin charging. These residues are not conserved in Uba1 and consistent with the FAT10-capability of UBA6.

Using the UBA6-FAT10 structure, the authors also attempt to explain the dual specificity of UBA6 for FAT10 and ubiquitin. A model of UBA6 in complex with ubiquitin was generated from a superimposed structure of the previously available UBA1-Ub complex. The authors identify UBA6 residues within the adenylation pocket that recognize the presence of distinct residues in ubiquitin and FAT10 (Arg42/Arg72 and Ile13/Tyr161, respectively). Single point mutations of Asp616Ala and Glu601Gln in UBA6 were designed to neutralize the charged residue that facilitates ubiquitin

recognition. Indeed, these mutations weakened the UBA6 charging reaction with ubiquitin (to 22% activity) while having no impact on FAT10 charging.

The authors suggest that these two sets of mutations, which selectively disable either FAT10 or ubiquitin activation by UBA6, will be useful for future studies in cells to probe the role of UBA6 in cellular signaling pathways. In order to make this claim, the authors should show evidence that these mutation do not disrupt charging of the E2 (USE1), which would disable the pathway regardless of distinguishing between ubiquitin and FAT10.

Overall, this is a valuable structural investigation of how UBA6 achieves dual specificity for ubiquitin and FAT10. The identification of mutations that selectively disable UBA6 functions will be of interest to the broad readership of Nature Communications.

Major comments:

1. Paragraph 2 of the introduction is poorly organized. Specifically, the argument that UBA6 is not a back-up system for UBA1.

2. The rationale for using the UBA6 chimera is poorly explained in the text. If the authors tried to crystallize wild-type UBA6 or tried other chimeras it should be explicitly stated.

Unless I missed this point, is there any comment or comparison of SCCH-UBA6 and SCCH-UBA1 domains between the reported UBA6-wt and UBA6-chimeric structures?

3. While discussing their results, the authors do not include enough references to the Figure panels they are discussing which makes it difficult to determine which Figures contain the results referred to in the text.

4. The authors comment on the fact that there are two bands in their FAT10ylation assay and refer to an experiment where DTT was added and one band disappeared. I could not locate this data in the manuscript but I think the results should be included in the manuscript.

5. When discussing the E601Q mutation in the paragraph starting on Line 502, the authors provide a rationale for why this mutation will disrupt ubiquitylation, but then say the effect is surprising in the next sentence. The authors should rework this section to improve clarity.

Minor comments:

1. The abstract is quite long.
2. The authors should add where the affinity tag cleavage sites are for each protein within the methods.
3. Figure 2 panels are not arranged logically (A-C in one column then D-F in another). The current arrangement gives the impression A relates to D (and so on).
4. In Figure 4, the authors should show the backbone of FAT10 as a colored ribbon to make it clear which protein contributes these amino acids (since the side chains aren't coloured).
5. Line 404: The authors say they superimpose UBA6-ATP with UBA6-FAT10 in Figure 4C. Should that be UBA1-Ub with UBA6-FAT10 (because Ub is in the Figure?)
6. On Line 449 the authors mention conservation of E601. It would be nice if the authors could also comment on whether the Histidines are present in UBA1?
7. Line 495: Typo D590 should be D591.
8. Throughout the manuscript, the authors refer to both D615A and D616A. Whichever is a typo should be corrected.
9. I think references in the text to Figure 4E are meant to be Figure 4F and vice-versa
10. Figure 6 panels and legend does not match (B and C).

Reviewer #2 (Remarks to the Author):

In this manuscript Truongvan et al. reports the crystal structure UBA6 in complex with FAT10, an unusual E1 activating enzyme responsible for the activation of the double-headed FAT10 modifier (FAT10ylation). Interestingly, UBA6 is also an E1 for ubiquitin. The authors also report the structure of yeast UBA1 bound to ATP at high resolution. The authors nicely describe the ATP binding pocket, which is basically conserved with UBA1, and the binding of double-headed FAT10, unexpectedly revealing the binding of the N-terminal FAT10 domain with a 3-helix bundle domain of UBA6. This novel region of UBA6 seems specific for binding to the double UbL-modifier FAT10 by means of a basic interface with the N-terminal FAT10 domain. The results presented in this manuscript are very

interesting, they increase the knowledge of the E1-activating enzyme functions, in this case in the activation of FAT10. The structural analysis is very well done, and the flow of the manuscript is very understandable. I recommend publication, and don't see any major issues that should be addressed, in my view.

Just a few corrections:

In Fig 1d, the C-terminal region of ISG15 is incorrect... I think that it should be similar to ubiquitin (LRLRGG). The authors have used the unmaturred ISG15 form and the alignment should be corrected.

In Fig 2, please indicate better the location of the crossover loop in the panels (color...). They are difficult to observe.

Perhaps it would be nice to show a clear figure of the components in the asymmetric unit and their contacts (UBA6-FAT10 and UBA6 empty).

Line 445. I guess that Fig 4f should be Fig 4e.

REVIEWERS' COMMENTS

Reviewer #1 (Remarks to the Author):

This manuscript describes crystal structures of UBA6 in complex with ATP and with FAT10. In order to solve a structure of UBA6 with FAT10, the authors use a chimeric version of UBA6 with the SCCH domain substituted from UBA1, the related but Ubiquitin-specific E1.

First, an ATP-bound UBA6 structure is compared with Uba1-ATP to reveal the mode of ATP interaction and hydrolysis is conserved (not surprising).

More interestingly, the UBA6-FAT10 complex provides a platform for understanding the ability of UBA6 to adenylate two different ubiquitin-like proteins, ubiquitin (Ub) and FAT10.

The structure of UBA6-FAT10 reveals the C-terminal FAT10 Ubl domain is positioned in the adenylation active site (similar to ubiquitin in a Uba1 complex) while the N-terminal FAT10 Ubl domain makes additional contacts with the 3HB (three helix bundle) of UBA6. A triple point mutation within the 3HB domain reduced UBA6 charging by FAT10 but had no effect on ubiquitin charging. These residues are not conserved in Uba1 and consistent with the FAT10-capability of UBA6.

Using the UBA6-FAT10 structure, the authors also attempt to explain the dual specificity of UBA6 for FAT10 and ubiquitin. A model of UBA6 in complex with ubiquitin was generated from a superimposed structure of the previously available UBA1-Ub complex. The authors identify UBA6 residues within the adenylation pocket that recognize the presence of distinct residues in ubiquitin and FAT10 (Arg42/Arg72 and Ile13/Tyr161, respectively). Single point mutations of Asp616Ala and Glu601Gln in UBA6 were designed to neutralize the charged residue that facilitates ubiquitin recognition. Indeed, these mutations weakened the UBA6 charging reaction with ubiquitin (to 22% activity) while having no impact on FAT10 charging.

The authors suggest that these two sets of mutations, which selectively disable either FAT10 or ubiquitin activation by UBA6, will be useful for future studies in cells to probe the role of UBA6 in cellular signaling pathways. In order to make this claim, the authors should show evidence that these mutations do not disrupt charging of the E2 (USE1), which would disable the pathway regardless of distinguishing between ubiquitin and FAT10.

Overall, this is a valuable structural investigation of how UBA6 achieves dual specificity for ubiquitin and FAT10. The identification of mutations that selectively disable UBA6 functions will be of interest to the broad readership of Nature Communications.

We highly appreciate the positive assessment of our study and thank the reviewer for his/her constructive criticisms.

Major comments:

1. Paragraph 2 of the introduction is poorly organized. Specifically, the argument that UBA6 is not a back-up system for UBA1.

The paragraph in question has been extensively rewritten and has also been shortened (see lines 53-66 of the revised manuscript).

2. The rationale for using the UBA6 chimera is poorly explained in the text. If the authors tried to crystallize wild-type UBA6 or tried other chimeras it should be explicitly stated. Unless I missed this point, is there any comment or comparison of SCCH-UBA6 and SCCH-UBA1 domains between the reported UBA6-wt and UBA6-chimeric structures?

Initially, we focused on the crystallization of the UBA6 wild-type in complex with FAT10. While we were able to obtain crystals, however, they showed only very poor diffraction. We then resorted to the chimeric construct in which the SCCH domain of UBA6 had been replaced with its counterpart from UBA1, which not only crystallized but also led to the structure presented in this study. This and other chimeric constructs (not discussed in the manuscript) were available to us since we had constructed them earlier to study how domain exchanges between UBA1 and UBA6 affect FAT10ylation. These experiments did not yield significant insights and hence are not included in our study.

In the revised version (Supplementary Fig. 4d) we included a superimposition of the SCCH domain of our UBA6 chimera, the SCCH domain of human UBA1 (PDB entry 6dc6) and the SCCH domain of yeast Uba1 (PDB entry 6zqh). As expected, the first two structures are more closely related to each other than to their yeast counterpart, however, they are not as similar as one might have expected given that they are after all identical in sequence. These diverging regions are located at the distal end of the SCCH and reflect the inherent conformational flexibility of this part of the SCCH, which also engages in contact with the E2 enzymes. A discussion of the structural differences was already present in the original manuscript and can now be found in a modified form in lines 396-402 of the revised manuscript. The section in question is also pasted here: “While the SCCH domains of UBA6_{chim} and its counterpart in UBA1 and Uba1 adopt quite similar structures in isolation (Supplementary Fig. 4d), the SCCH in UBA6_{chim} undergoes drastic conformational changes. Compared to the Uba1-ubiquitin complex the SCCH is rotated by more than 60° towards the UFD coupled to a displacement of the crossover loop by up to 10 Å near the active site cysteine. These conformational changes are observed in both copies of UBA6_{chim} which exhibit almost identical structures (Supplementary Fig. 4e) and hence do not seem to be induced by FAT10 binding and may be due to the chimeric nature of this construct.”

3. While discussing their results, the authors do not include enough references to the Figure panels they are discussing which makes it difficult to determine which Figures contain the results referred to in the text.

We apologize for this oversight and have included additional references to the figures, or, more precisely, to the respective panel being discussed in the text.

4. The authors comment on the fact that there are two bands in their FAT10ylation assay and refer to an experiment where DTT was added and one band disappeared. I could not locate this data in the manuscript but I think the results should be included in the manuscript.

The corresponding data were in fact not shown and we therefore should have at least added “data not shown” to avoid any confusion. We are grateful for the suggestion to include these data which are now shown in Supplementary Fig. 4c.

5. When discussing the E601Q mutation in the paragraph starting on Line 502, the authors provide a rationale for why this mutation will disrupt ubiquitylation, but then say the effect is surprising in the next sentence. The authors should rework this section to improve clarity.

The section in question has been revised (see lines 494-505).

Minor comments:

1. The abstract is quite long.

The abstract has been shortened by ~40% and now contains 148 words.

2. The authors should add where the affinity tag cleavage sites are for each protein within the methods.

The information has been added in the Methods section, subheading Molecular Biology, as requested.

3. Figure 2 panels are not arranged logically (A-C in one column then D-F in another). The current arrangement gives the impression A relates to D (and so on).

Figure 2 was rearranged as suggested with panels a - c in the top row and panels d - f in the bottom row.

4. In Figure 4, the authors should show the backbone of FAT10 as a colored ribbon to make it clear which protein contributes these amino acids (since the side chains aren't coloured).

Figure 4 has been modified so that the C-atoms in the main chain are now colored in yellow.

5. Line 404: The authors say they superimpose UBA6-ATP with UBA6-FAT10 in Figure 4C. Should that be UBA1-Ub with UBA6-FAT10 (because Ub is in the Figure?)

The reviewer is correct, thank you for spotting this mistake.

6. On Line 449 the authors mention conservation of E601. It would be nice if the authors could also comment on whether the Histidines are present in UBA1?

The histidines are not conserved in UBA1 where they are replaced by Asn and Ser. This information is now presented in line 439 and can also be derived from Fig. 5b.

7. Line 495: Typo D590 should be D591.

Thank you for spotting this error, which has been corrected.

8. Throughout the manuscript, the authors refer to both D615A and D616A. Whichever is a typo should be corrected.

The correct number is 616 and the incorrect D615A descriptors have been corrected. Again, we thank the reviewer for spotting these mistakes.

9. I think references in the text to Figure 4E are meant to be Figure 4F and vice-versa

The references to Fig. 4e and 4f were indeed incorrect and this has been fixed.

10. Figure 6 panels and legend does not match (B and C).

This mismatch has been corrected. Thank you for pointing this out.

Reviewer #2 (Remarks to the Author):

In this manuscript Truongvan et al. reports the crystal structure UBA6 in complex with FAT10, an unusual E1 activating enzyme responsible for the activation of the double-headed FAT10 modifier (FAT10ylation). Interestingly, UBA6 is also an E1 for ubiquitin. The authors also report the structure of yeast UBA1 bound to ATP at high resolution. The authors nicely describe the ATP binding pocket, which is basically conserved with UBA1, and the binding of double-headed FAT10, unexpectedly revealing the binding of the N-terminal FAT10 domain with a 3-helix bundle domain of UBA6. This novel region of UBA6 seems specific for binding to the double UbL-modifier FAT10 by means of a basic interface with the N-terminal FAT10 domain. The results presented in this manuscript are very interesting, they increase the knowledge of the E1-activating enzyme functions, in this case in the activation of FAT10. The structural analysis is very well done, and the flow of the manuscript is very understandable. I recommend publication, and don't see any major issues that should be addressed, in my view.

Again, we highly appreciate the positive assessment of our study and thank the reviewer for his/her efforts in improving our manuscript.

Just a few corrections:

1. In Fig 1d, the C-terminal region of ISG15 is incorrect... I think that it should be similar to ubiquitin (LRLRGG). The authors have used the unmaturing ISG15 form and the alignment should be corrected.

The reviewer is correct, these were the C-terminal residues for the immature form of ISG15 and this error has been corrected. The last six residues are indeed identical in the processed forms of ISG15 and ubiquitin.

2. In Fig 2, please indicate better the location of the crossover loop in the panels (color...). They are difficult to observe.

We presume the reviewer was referring to Fig. 1 since the crossover and reentry loop were not shown in Fig. 2. We have added arrows to panels a-c in Fig. 1 to better indicate the location of the crossover and reentry loops.

3. Perhaps it would be nice to show a clear figure of the components in the asymmetric unit and their contacts (UBA6-FAT10 and UBA6 empty).

We thank the reviewer for this suggestion and have added two panels (a and b) to Supplementary Fig. 5 which illustrate the content of the asymmetric unit in the C2 and P1 unit cells, respectively. Panel b also features another UBA6 molecule to illustrate how the sole FAT10 moiety in the cell is contacted by the two UBA6 molecules.

4. Line 445. I guess that Fig 4f should be Fig 4e.

We thank the reviewer for spotting this mistake which has been corrected.